# Comparative Profiling of Volatile Compounds and Fatty Acids in Pomegranate Seed Oil: Soxhlet vs. CO_2_/IPA Extraction for Quality and Circular Bioeconomy Goals

**DOI:** 10.3390/foods14172951

**Published:** 2025-08-25

**Authors:** Caterina Fraschetti, Antonello Filippi, Antonia Iazzetti, Giancarlo Fabrizi, Francesco Cairone, Stefania Cesa

**Affiliations:** 1Department of Chemistry and Technologies of Drug, University “La Sapienza” of Rome, P.le Aldo Moro 5, 00185 Rome, Italy; caterina.fraschetti@uniroma1.it (C.F.); antonello.filippi@uniroma1.it (A.F.); giancarlo.fabrizi@uniroma1.it (G.F.); stefania.cesa@uniroma1.it (S.C.); 2Department of Basic Biotechnological Sciences, Intensivological and Perioperative Clinics, Catholic University of the Sacred Heart, L.go F. Vito 1, 00168 Rome, Italy; antonia.iazzetti@unicatt.it

**Keywords:** pomegranate seed oil, supercritical CO_2_ extraction, punicic acid, HS-SPME-GC-MS, NMR, unsaturated fatty acids

## Abstract

This study compares the chemical profiles of pomegranate seed oil (PSO) from two cultivars, Granato (G) and Roce (R), extracted by Soxhlet and supercritical CO_2_/isopropanol. GC-MS and NMR analyses confirmed punicic acid as the dominant fatty acid, with α-eleostearic, oleic, and linoleic acids in lower amounts. Supercritical extraction increased yield (about 18%) and selectively raised α-eleostearic and linoleic acids. Volatile organic compound (VOC) profiling by HS-SPME-GC-MS showed higher aldehydes, esters, and terpenes in supercritical extracts, including (E)-cinnamaldehyde (absent in Soxhlet). Soxhlet oils contained more hydrocarbons, suggesting thermal degradation. Overall, supercritical CO_2_/IPA proved more sustainable and selective, preserving nutritional and aromatic quality and supporting PSO’s potential in food, nutraceutical, and cosmetic uses.

## 1. Introduction

The increasing demand for sustainable production systems has prompted both scientific and industrial communities to reassess the value of agri-food by-products within the framework of a circular economy [1]. This model seeks to minimize waste generation and promote the valorization of residual biomass as high-value-added resources, thereby contributing to a reduction in the overall environmental footprint [2,3]. According to FAO estimates, fruit and vegetable waste in industrialized countries amounts to approximately USD 680 billion annually, representing a significant fraction of global organic waste [2].

Among agri-food by-products, pomegranate (*Punica granatum* L.) has attracted considerable attention due to its high content of bioactive compounds, particularly in the peel and seeds, which can constitute up to 50% of the total fruit weight and are typically discarded during juice production [4,5]. Pomegranate seeds, besides other interesting characters, are a rich source of oil with a distinctive lipid profile dominated by punicic acid, a conjugated trienoic fatty acid from the omega-5 family. Punicic acid (9Z,11E,13Z-octadecatrienoic acid), an isomer of α-linolenic acid, can account for up to 80% of the total fatty acids in pomegranate seed oil (PSO). This fatty acid has been associated with antioxidant, anti-inflammatory, and potential anticancer effects [6,7]. Emerging evidence suggests its role in modulating lipid and glucose metabolism, reducing inflammatory markers, and preserving endothelial function, indicating its relevance in the prevention of metabolic and cardiovascular disorders [8]. In addition to punicic acid, PSO contains other unsaturated fatty acids such as oleic, linoleic, catalpic, and α-eleostearic acids, which may exert synergistic biological effects. The composition and relative abundance of these fatty acids are influenced by multiple variables including cultivar, seed maturity, and extraction technique [9]. The PSO can be obtained through conventional methods, such as Soxhlet extraction with non-polar solvents, and innovative approaches like supercritical CO_2_ (SFE). Soxhlet offers high lipid recovery but requires large solvent volumes, prolonged heating, and may degrade thermolabile compounds [7,10]. In contrast, SFE, especially with polar co-solvents such as isopropanol, operates under milder, solvent-free conditions and better preserves bioactive molecules [11]. These methods are widely applied to other vegetable oils, such as Soxhlet in soybean, sunflower, and flaxseed, and SFE in olive, grape seed, chia, and hemp oils, where it improves yield, nutritional quality, and aromatic complexity [12,13].

Therefore, detailed fatty acid profiling not only makes nutritional and functional assessments possible but also serves as a tool for product traceability and quality standardization. Despite the growing interest in the functional properties of PSO, its aromatic profile remains largely underexplored. The headspace fraction, comprising volatile organic compounds (VOCs), plays a critical role in defining sensory characteristics and may serve as a chemical signature for cultivar and processing identification. As in the case of extra virgin olive oil (EVOO), where the aroma profile is a key marker of varietal authenticity, technological history, and quality, the volatile composition of PSO holds potential as an indicator of origin and processing conditions [14]. In extra virgin olive oils (EVOOs), the volatile profile is predominantly shaped by enzymatic oxidation of unsaturated fatty acids via the lipoxygenase (LOX) pathway, generating a mixture of aldehydes, alcohols, ketones, esters, and hydrocarbons. These compounds not only contribute to sensory perception but also exhibit antioxidant and antimicrobial activities [15]. A similar mechanistic pathway could be hypothesized for PSO, particularly due to its high content of conjugated polyunsaturated fatty acids, which are susceptible to oxidative and enzymatic transformation into bioactive volatiles.

In this context, the present study aims to use HS-SPME-GC-MS for characterizing the headspace volatile profile of pomegranate seed oil extracted from two cultivars (*Punica granatum* L., Roce and Granato) using two extraction methods (Soxhlet and CO_2_/isopropanol supercritical system). Concurrently, the unsaturated fatty acid composition of the oils has been analyzed through NMR and GC-MS to explore potential correlations between lipid content and volatile expression. This integrated analytical approach is intended to generate a comprehensive chemical fingerprint of PSO, supporting the identification of quality markers for cultivar differentiation, product standardization, and valorization. Ultimately, this study contributes to the development of analytical tools for the classification and authentication of functional oils, aligned with circular economy principles and sustainable agri-food resource management.

## 2. Materials and Methods

### 2.1. Sample Preparation

Dried pomegranate seeds from cultivars Granato (G) and Roce (R) were kindly gifted by the biological company Giovomel (Avellino, Italia). The received seeds were then finely homogenized by a domestic mixer (Bimby Thermomix TM31, Vorwerk, Wuppertal, Germany).

### 2.2. Soxhlet Extraction with N-Hexane

Oil extraction using the Soxhlet method was performed as described in the previous study by Cairone et al., 2023 [6]. Approximately 20 g of finely ground pomegranate seeds were subjected to Soxhlet extraction using a 500 mL apparatus and *N*-hexane (500 mL) as the solvent. The extraction was carried out for 4 h at 80 °C. Following extraction, the solvent was removed under reduced pressure using a rotary evaporator. The resulting oil (R1 and G1) were stored at 4 °C until subsequent analyses.

### 2.3. CO_2_/Isopropanol (IPA) Supercritical Extraction

Supercritical CO_2_ extraction was conducted using a dedicated system (Jasco Europe s.r.l., Cremella, Italy), with methodological adjustments based on Cairone et al., 2023 [6]. The apparatus included a Jasco-PU-4347 scCO_2_ pump (Jasco Europe s.r.l., Cremella, Italy) connected to a stainless-steel extraction vessel (10 mL), which was equipped with 10 µm porous sintered filter disks (316 L grade). Approximately 4.0 g of homogenized pomegranate seeds was loaded into the vessel, which was maintained at 50 °C using a Jasco CO-4065 oven (Jasco Europe s.r.l., Cremella, Italy). The target pressure was achieved by adjusting the CO_2_-BP-4390 back pressure regulator. Following a 30 min equilibration phase, the extraction was carried out using a CO_2_/isopropanol mixture (90:10 *v*/*v*) at 40 MPa, with a solvent flow rate of 5 mL/min for 2 h. The resulting extract was concentrated under reduced pressure using a rotary evaporator (samples R2 and G2), and the oils were stored at 4 °C until analysis.

### 2.4. Transesterification Procedure

In a reactor equipped with a magnetic stir bar, 200.0 mg of oil was mixed with 0.4 mL of methanol. Sodium methoxide (48.2 mg, 0.9 mmol) was then added to the mixture, which was stirred at room temperature for 3 h. Finally, the reaction mixture was diluted with ethyl acetate (50 mL) and washed three times with water (3 × 40 mL). The combined organic phases were dried over anhydrous sodium sulfate, filtered, and evaporated under reduced pressure to yield pale-yellow transesterified products (R1_FAME_, G1_FAME_, R2_FAME_, and G2_FAME_).

### 2.5. ^1^H and ^13^C-NMR Analysis

Aliquots of 0.20 mL from each transesterified oil sample (R1_FAME_, G1_FAME_, R2_FAME_ and G2_FAME_) were dissolved in 0.5 mL of deuterated chloroform (CDCl_3_, isotopic enrichment ≥ 99.80%) and transferred into NMR tubes for analysis. ^1^H and ^13^C NMR spectra were recorded on a Bruker Avance 400 spectrometer (400.13 MHz for ^1^H and 100.6 MHz for ^13^C) equipped with a Nanobay console and a Cryoprobe Prodigy probe (Bruker, Milan, Italy). Quantitative ^13^C NMR spectra were recorded using inverse-gated decoupling (pulse program *zgig*). ^1^H NMR spectra were acquired with a relaxation delay (D1) of 1.0 s and an acquisition time (AQ) of 4.090 s, resulting in a total recycle time of 5.090 s. ^13^C NMR spectra were recorded with D1 = 2.0 s and AQ = 1.363 s (recycle time = 3.363 s). The acquired spectra were processed and analyzed using Bruker TopSpin 3.5.2 software .

### 2.6. GC-MS and HS-SPME/GC-MS Analyses

The preparation of fatty acid methyl ester (FAME) stock solutions was performed by dissolving the individual R1_FAME_, G1_FAME_, R2_FAME_, and G2_FAME_ sample in deuterated chloroform (CDCl_3_) to a final 80 mg/mL concentration. These solutions were subsequently diluted with cyclohexane to achieve a final concentration of 1 mg/mL to be submitted to the gas chromatographic analysis. Approximately 1 μL of each diluted sample was directly injected into the GC inlet. For volatile organic compound (VOC) analysis, 40 mg aliquots of the R1, G1, R2, and G2 oil samples were transferred in 4 mL vials, which were placed in a thermostatic bath and allowed to equilibrate at 70 °C for 20 min. Then, the DVB-CAR-PDMS fiber, selected based on its broad analyte selectivity, was exposed to the headspace of each vial at 70 °C for 20 min to facilitate volatile compound extraction. The fiber was finally introduced into the inlet of a gas chromatograph. All the analyses were performed in triplicate in a gas chromatograph (6850, Agilent Technologies, Santa Clara, CA, USA) coupled with a mass spectrometer (5975, Agilent Technologies, Santa Clara, CA, USA). The GC parameters were set as follows: column, HP-5ms (30 m × 0.25 mm × 0.25 um); injector temperature, 250 °C (260 °C for the HS-SPME technique); temperature program, 40 °C maintained for 5 min, temperature rise (5 °C/min) to 200 °C, kept for 20 min. Mass spectrometry parameters were set as follows: EI energy, 70 eV; source temperature, 230 °C; quadrupole temperature, 150 °C.

For both the HS-SPME-GC-MS and direct GC-MS analyses, mass spectrometric and chromatographic data were combined to confirm analyte identification. Electron ionization (EI) spectra were matched against reference spectra from commercial databases (FFNSC 3) and freely available online resources (NIST 11, Flavor2). Kovats retention indices (KIs) served as an additional confirmation parameter for the mass spectrometry-based identifications. The KI values were determined using an *N*-alkane standard mixture (C7–C40) under identical chromatographic conditions and subsequently compared with database values and published literature data [16]. Semi-quantitative analysis involved the automatic integration of all GC-MS peaks with S/N ratios exceeding 3, and the results were expressed as area percentages of the total chromatographic area.

### 2.7. Statistical Analysis

Each assay was performed in triplicate or more, and results are expressed as mean ± standard deviation (SD). Statistical analysis and graph layout were carried out using one-way ANOVA followed by Tukey’s Honest Significant Difference (HSD) post-hoc test for multiple comparisons in XLSTAT 2021.4 software (New York, NY, USA). Differences were considered statistically significant at *p* < 0.05, with *p*-values reported to assess reproducibility and significance across replicates.

## 3. Results and Discussion

### 3.1. Extraction Procedure

The extraction method is a key factor influencing both the quantitative yield and the qualitative composition of PSO. The freshly extracted pomegranate seed oils presented a clear, transparent appearance, with low turbidity and absence of suspended particles. All samples displayed a fluid consistency typical of unsaturated oils, without signs of phase separation or sediment. The aroma was faint but distinctive, with mild nutty and fruity notes more pronounced in the CO_2_/IPA extracts, in agreement with their richer volatile fraction. The scCO_2_ method demonstrates superior energy efficiency compared to conventional Soxhlet extraction, as evidenced by previous studies [6]. While Soxhlet extraction typically necessitates 4–6 h at temperatures ranging from 69–90 °C, utilizing substantial volumes of *N*-hexane and incurring additional energy for solvent evaporation and recovery, scCO_2_ extraction achieves comparable yields in a mere 2 h at 50 °C and 40 MPa. This is accomplished through the use of CO_2_ and a minimal quantity of isopropanol as a co-solvent, both of which are readily recoverable and reusable.

These findings align with the existing literature, where scCO_2_ is widely recognized for its capacity to reduce processing time, operational temperature, and overall energy consumption in comparison to traditional solvent-based techniques [17,18]. From an industrial perspective, the elevated capital investment associated with high-pressure equipment must be acknowledged [19]. However, for the production of high-value products such as nutraceuticals, functional ingredients, and cosmetics, the diminished energy footprint, the production of solvent-free extracts, and the enhanced product quality render scCO_2_ extraction a more sustainable and economically competitive alternative to Soxhlet.

So, in this study, we compared two extraction strategies: a conventional Soxhlet extraction with *N*-hexane (samples G1 and R1), and a supercritical CO_2_ extraction co-solvated with isopropanol (90:10 *v*/*v*) (samples G2 and R2). To better explain the relative differences between the two extraction procedures, a radar plot is reported in Figure 1.

The efficiency of the two extraction methods was first evaluated in terms of oil yield (Figure 1, Panel (B)). Soxhlet extraction was carried out using 20 g of ground pomegranate seeds and 500 mL of *N*-hexane at 80 °C for 4 h, whereas the CO_2_/IPA method was performed using 4 g of seed material, operating at 50 °C and 40 MPa for 2 h, with significantly reduced solvent volume and thermal exposure. Despite these milder conditions, CO_2_/IPA consistently produced higher yields: 17.7% for G2 and 16.3% for R2, compared to 12.9% for G1 and 14.6% for R1 using Soxhlet extraction.

This difference is clearly visualized in the radar plot (Figure 1, Panel (A)), which illustrates the superior performance and consistency of the CO_2_/IPA method across both cultivars. A comparison of extraction yields, together with available energy consumption data, suggests that one method may offer greater energy-saving potential than the other, indicating superior eco-efficiency under the tested conditions [20]. These findings are consistent with literature values reported for both conventional and unconventional extraction technologies. Soxhlet extractions typically yield around 14–15% when petroleum ether is used [7,21], whereas cold pressing produces 5–10% and ultrasound-assisted ethanol extraction gives 8–13%. In this context, the CO_2_/IPA yields fall at the upper end of the expected range, confirming the effectiveness of the supercritical approach [11,12].

In addition to oil recovery, this study placed emphasis on the more eco-friendly profile and on the method’s selectivity for bioactive compounds. The use of 10% isopropanol as a co-solvent likely improved the solubility of moderately polar lipids, thereby enhancing extraction efficiency without compromising sustainability. Conversely, Soxhlet extraction involves the use of large volumes of toxic solvent (500 mL of hexane) and prolonged heating, raising concerns related to scalability, safety, and environmental impact [10].

Beyond yield comparisons, this work explores how extraction methodology affects the chemical composition of the oils, particularly the isomeric distribution of conjugated linolenic acids (CLnAs), notably punicic, α-eleostearic, and catalpic acids, which are crucial bioactive lipids in PSO. By integrating GC-MS and NMR profiling, we investigated how each method influences both the recovery and geometric configuration of these fatty acid isomers, alongside other unsaturated compounds [6].

### 3.2. NMR Analysis

In our previous study [6], the application of ^1^H and ^13^C NMR spectroscopy to transesterified pomegranate seed oil provided one of the first detailed characterizations of the isomeric distribution of conjugated linolenic acids (CLNAs) in *Punica granatum* L. oils. Although the cultivars (Granato and Roce) were the same, the seeds were harvested in different crop years. That study identified punicic acid (cis-9, trans-11, cis-13) as the predominant isomer, accompanied by α-eleostearic, catalpic, and trace levels of β-eleostearic acid. Notably, the Soxhlet-extracted oils showed a broader isomeric diversity, likely due to thermal isomerization induced by prolonged solvent reflux ate relatively high temperature (80 °C, for 4 h).

In the present work, similar NMR methods were employed following transesterification, with particular attention to the diagnostic olefinic region (125–136 ppm) of the ^13^C inverse-gated spectra [6,22,23]. The same core CLNA isomers were observed in all samples (^13^C-NMR spectra are reported in Appendix A) and the quali-quantitative identification of the acyl fingerprint was carried out according to the method reported in our previous work. However, differences in their relative distribution were evident, especially in the Roce cultivar. The quantitative data (Table 1) clearly show that punicic acid remained the most abundant isomer across all extractions, with G1_FAME_ and G2_FAME_ both exhibiting 85% of this fatty acid, in the acyl composition. In contrast, R2_FAME_ contained about 80% punicic acid, even lower than its Soxhlet counterpart R1_FAME_ (82%).

These differences, also confirmed by GC-MS, suggest that while the supercritical extraction method effectively preserves punicic acid in the ‘Granato’ cultivar, its efficiency appears lower in ‘Roce’. This may be due to cultivar-specific susceptibility to isomerization or oxidative transformations under supercritical conditions, a phenomenon broadly discussed by Avato and Tava (2022) [13].

Furthermore, the relative increase of α-eleostearic acid from 3% in R1_FAME_ to 5% in R2_FAME_ may point to a possible influence of the supercritical method on the isomeric profile, potentially favoring the extraction of this compound in the Roce cultivar. Catalpic acid was only detected in G1_FAME_ (1.4%), highlighting a potential method- and cultivar-dependent extraction pattern. These minor isomers are biologically relevant, as they exhibit different antioxidant and cytotoxic effects, with structural configuration influencing their metabolic activity and oxidative stability [24].

In addition to CLNAs, a higher relative content of monounsaturated (oleic) and diunsaturated (linoleic) fatty acids was observed in CO_2_/IPA samples. This effect is more pronounced in R2_FAME_, which shows about 9% of oleic and linoleic acid, compared to 7–8% in R1_FAME_. However, this apparent increase may not reflect enhanced extraction of these compounds, but rather a relative shift resulting from the reduced punicic acid content under supercritical conditions. This interpretation is consistent with the overall compositional changes and highlights the importance of evaluating fatty acid profiles in relative rather than absolute terms [25,26].

These results demonstrated that extraction method significantly influences the acyl profile of PSO, not only in terms of yield but also in the relative abundance and preservation of functionally relevant isomers. Soxhlet extraction appears to promote greater isomer diversification, likely due to thermal rearrangements, as previously reported in linolenic acid-rich oils processed under harsh conditions [27]. Conversely, supercritical CO_2_/IPA extraction exhibits greater selectivity for native CLNA configurations and allows for better recovery of nutritionally valuable unsaturated fatty acids.

From a functional standpoint, these compositional differences have relevant practical implications. Preserving native punicic acid and maintaining specific isomeric profiles could significantly affect the oil’s oxidative stability, the consequent bioavailability, and therefore the nutritional properties. Consequently, beyond its environmental benefits, supercritical CO_2_/IPA extraction emerges as a chemically selective and cultivar-responsive method, promoting both quality and sustainability in the production of high-value pomegranate seed oil.

### 3.3. GC-MS Analysis of FAMEs Fraction

The detailed FAME composition of G1–G2 and R1–R2 extracts reported in Appendix A is summarized in Table 2 and graphically depicted in Figure 2. The most striking aspect of the GC-MS data is the overwhelming dominance of punicic acid, which comprises between 80.3% and 84.8% of the total fatty acid composition across all samples. Conjugated triene systems have been identified in different plants, wherein the most representative species is (9Z,11E,13E)-octadeca-9,11,13-trienoic acid (α-eleostearic acid) accompanied by its isomer (9E,11E,13E)-octadeca-9,11,13-trienoic acid (β-eleostearic acid) [13]. Punicic acid and (9Z,11E,13Z)-octadeca-9,11,13-trienoic acid (catalpic acid) are 9,11,13 trienoic geometrical isomers generally found in remarkable quantities in *Punica* and *Catalpa* seed oils [16].

From a biosynthetic perspective, the punicic acid dominating the fatty acid profile of pomegranate extracts tells something peculiar about the metabolic machinery of pomegranate seeds. The plant must possess highly active and specific desaturase enzymes capable of creating this conjugated system with remarkable efficiency [24]. While punicic acid dominates the profile, the minoritarian fatty acids provide equally important insights into the biological aspects of these extracts. The presence of 4.3–6.2% α-eleostearic acid is particularly intriguing from a biochemical standpoint. This isomer of punicic acid differs only in the geometry of the terminal double bond, yet this subtle difference can profoundly impact biological activity. The consistent ratio of punicic to α-eleostearic acid across the analyzed samples suggests the occurrence of a natural biosynthetic balance in pomegranate seeds. The oleic acid content, varying from 4.7% to 8.3%, provides another layer of complexity to consider. Oleic acid serves as the biosynthetic precursor to many polyunsaturated fatty acids, and its relative abundance in our samples might reflect the balance between different desaturase activities. The higher oleic acid levels in the R_FAME_ samples compared to the G_FAME_ ones could indicate differences in the timing of fatty acid synthesis during seed development, or in the efficiency of the downstream desaturase systems that convert oleic acid to more highly unsaturated products. This aspect is confirmed by the NMR analysis, as shown by the data reported in Table 1, which clearly illustrate the variations in signal intensities corresponding to olefinic protons (see Appendix A). These differences further support the hypothesis of altered desaturase activity or temporal shifts in fatty acid biosynthesis pathways between the R_FAME_ and G_FAME_ samples. The between-cultivar differences in fatty acid profiles may reflect genotype-related variations in desaturase/conjugase activity (e.g., FADX). In fact, the distinct fatty acid signatures observed between cultivars likely stem from inherent genetic differences influencing desaturase/conjugase pathways (notably FADX), the key enzymatic system for punicic acid biosynthesis. Such genotype-driven modulation of lipid metabolism has been documented in pomegranate and other oilseeds, where variation in FAD-type enzyme activity can markedly reshape the seed oil fatty acid profile [28,29].

Concerning the comparative analysis of the sample’s variations, the G1–2_FAME_ samples definitely exhibited a higher punicic acid content (84.7–84.8%) compared to the R1–2_FAME_ samples (80.3–81.1%), regardless of the extraction method. An inverse relationship between punicic acid and oleic acid (R_FAME_: 7.2–8.3% vs. G_FAME_: 4.7–5.0%) can be due to the biosynthetic pathway of punicic acid, whose indirect precursor is oleic acid itself [26]. Further evidence emerging form Figure 2 is the inverse relationship between punicic acid and α-eleostearic acid, which is more efficiently synthesized in the R1–2 samples. The compositional data obtained from GC-MS were found to be in strong agreement with the NMR findings, particularly for punicic acid, which consistently emerged as the dominant conjugated linolenic acid isomer across all samples. The relative proportions of α-eleostearic and oleic acids also showed comparable trends between the two analytical platforms, supporting the robustness and reproducibility of the isomeric profile regardless of the employed technique.

### 3.4. HS-SPME-GC-MS Analysis of VOC Fraction

The composition of the VOC fraction arising from samples G1/R1 and G2/R2 pairs of samples is reported in Table 3 and Table 4, respectively.

The HS-SPME-GC-MS analysis pointed to remarkable qualitative and quantitative differences in the VOC profiles of the four pomegranate extracts, namely G1, G2, R1, and R2. These differences were not only related to the intensity and distribution of individual compounds but also to the overall chemical classes represented in each sample. It is worth noting that the X1 (X = G,R) and X2 (X = G,R) pairs exhibited distinct compositional patterns, mainly due to the most representative class of compounds. This difference reflects a different selectivity of the extraction method (Figure 3 and Figure 4).

*Hydrocarbons: Alkanes, Cycloalkanes, and Alkenes.* Hydrocarbons constituted a remarkably populated class of VOCs in all samples, but with significantly different profiles across them. G1 and R1 were particularly rich in alkanes and cycloalkanes, which together accounted for a large proportion of the total chromatographic area (Figure 3 and Figure 4). In G1, alkanes such as nonane (1.89%), decane (3.59%), and undecane (1.34%) were prominent, alongside several unidentified alkanes at higher retention indices (e.g., RI 1115: 6.11%). Cycloalkanes were even more significant in G1 and R1, with 1,1′-bicyclohexyl emerging as the dominant compound in both samples (49.73–54.25%). Other cycloalkanes such as trans-decahydro-naphthalene and cis-1-methyl-4-(1-methylethyl)-cyclohexane were also observed and strongly highlighted by the radar plot in Figure 3 (Panel (B)). The overwhelming presence of 1,1′-bicyclohexyl in X1 (X = G,R) samples, which to our knowledge has never been reported as a natural component of seed oils, may suggest a potential contamination during the processing workflow. By contrast, G2 and R2 exhibited much lower hydrocarbon content (Figure 3 and Figure 4). Alkanes such as octane, nonane, and decane were detected but in relatively minor quantities (generally < 1%), suggesting a shift away from saturated hydrocarbon dominance. This points to a selective extraction of more polar or oxygenated volatiles in G2 and R2 (Figure 3, Panel (D)). Alkenes were present exclusively in R1, G2, and R2, with compounds such as 1-decene and 1-dodecene being notable.

*Aromatic Hydrocarbons: Alkylbenzenes and Derivatives.* Alkylbenzenes were present in all four extracts but were most prominent in G1, where ethylbenzene (1.45%), dimethylbenzene (10.11%), and mesitylene (1.43%) constituted significant portions of the VOC profile. These compounds are often derived from phenylpropanoid metabolism or thermal degradation processes and contribute to the characteristic aromatic signatures of some pomegranate cultivars [30].

*Aldehydes.* Aldehydes, which are nearly absent in G1 and <10% in R1, lie in the 15.89–57.43% range in the X2 (*n* = G, R) samples, suggesting once more an extraction method more selective for oxygenated compounds for the latter X2 samples. Aldehydes are particularly represented by (E)-cinnamaldehyde (32.89% in G2; 1.96% in R2) and nonadienals (11.55% in G2; 9.93% in R2), which are associated with sweet, spicy, and green aromas and are significant contributors to the flavor and aroma profile of pomegranate extracts. While 2,4-nonadienal is most likely produced during the oxidation of linoleic acid [31], the presence of (E)-cinnamaldehyde has not yet been reported in the literature.

*Terpenes and Sesquiterpenes.* Terpenoids represent a class of biologically active and aromatic compounds frequently associated with plant secondary metabolism. The only terpene-rich sample was G2, which contains a diverse array of mono- and sesquiterpenes, including α-pinene, β-pinene, myrcene, limonene, terpinolene, and the sesquiterpene E-caryophyllene. These compounds are commonly associated with fruity, piney, or herbal aromas, and their presence suggests its greater aromatic complexity. Notably, myrcene, α-pinene, and limonene were among the most abundant terpenes in the G2 extract.

From a compositional standpoint, the volatile fraction of the oils revealed significant differences between the two extraction methods. While the fatty acid profiles were relatively stable across all samples, the analysis of volatile organic compounds (VOCs) highlighted clear distinctions in both complexity and functional potential.

Soxhlet-extracted oils (G1, R1) were characterized by hydrocarbon-rich profiles, predominantly alkanes and cycloalkanes, which are commonly associated with solvent residues or thermal degradation by-products. This observation is consistent with previous data on pomegranate seed oil and with the literature describing the non-selective nature of Soxhlet extraction in targeting non-polar and potentially contaminant compounds under prolonged heating [6,32]. The possible thermal degradation observed in Soxhlet extracts is consistent with the literature showing that prolonged heating of highly unsaturated oils promotes oxidation/isomerization of PUFA and depletion of thermolabile VOCs with concurrent formation of lipid oxidation aldehydes (e.g., nonanal, 2,4-nonadienal). In pomegranate seed oil specifically, punicic acid is reported to be susceptible to isomerization under thermal/oxidative stress (and even during acid-catalyzed derivatization at ≤90 °C), while heating studies on vegetable oils consistently document loss of oxygenated volatiles and build-up of oxidation markers under thermal load [33,34].

Conversely, the CO_2_/IPA-extracted oils (G2, R2) exhibited a markedly different VOC profile, enriched in oxygenated species such as aldehydes, alcohols, esters, and terpenes. These classes of volatiles are known for their aromatic contribution and bioactivity and suggest that the supercritical extraction not only preserves but may also selectively enhance the recovery of thermolabile and functional molecules. This pattern aligns with previous studies demonstrating that supercritical CO_2_, especially when combined with a polar co-solvent like isopropanol, can extract bioactive volatile fractions more efficiently and with lower degradation risk [35,36].

Of particular interest is the detection of (E)-cinnamaldehyde exclusively in the CO_2_/IPA samples. This compound, known for its antioxidant, antimicrobial, and anti-inflammatory properties, further underscores the functional advantages of this extraction method [37,38]. Overall, these findings support the idea that CO_2_/IPA extraction not only improves oil yield and fatty acid preservation but also promotes the selective recovery of volatile bioactives, thereby enhancing the nutritional and sensory value of pomegranate seed oil.

## 4. Conclusions

This study demonstrates that the choice of extraction method significantly influences not only the overall yield but also the chemical fingerprint of pomegranate seed oil, both in the fatty acid profile and in the volatile composition. Supercritical CO_2_/IPA extraction consistently outperformed conventional Soxhlet extraction in terms of efficiency, sustainability, and compositional quality.

From a green chemistry perspective, the CO_2_/IPA method required lower solvent volumes, milder operating conditions, and resulted in higher extraction yields (16.5–17.5% compared to 13–14.5%), while also preserving several bioactive compounds such as (E)-cinnamaldehyde, phenylethanol, and terpenes. These findings underscore its potential as a sustainable and selective extraction approach for the valorization of pomegranate by-products. GC-MS and NMR analyses were in strong agreement, confirming punicic acid as the dominant conjugated linolenic acid in all extracts (~83%), accompanied by lower amounts of α-eleostearic acid (4–6%), oleic acid (5–8%), and linoleic acid (5–9% by NMR, ~3% by GC-MS). Notably, the supercritical extraction method increased the relative abundance of α-eleostearic and linoleic acids in the Roce cultivar, suggesting an interplay between extraction conditions and genotype-specific lipid biosynthetic pathways.

Importantly, the PCA analysis (Figure 5) integrating volatile and fatty acid data highlighted a clear compositional segregation between the Soxhlet and supercritical CO_2_/IPA-extracted samples. Soxhlet-derived oils (G1, R1) clustered closely due to their saturated hydrocarbon content and reduced aromatic complexity, while CO_2_/IPA-extracted oils (G2, R2) formed a separate group driven by their richer and more diverse oxygenated volatile profile and bioactive lipid content. This differentiation reinforces the idea that supercritical CO_2_/IPA extraction not only enhances yield and purity but also tailors the chemical profile of PSO toward higher nutritional and functional value.

In summary, this integrated approach confirms that supercritical CO_2_/IPA extraction represents a highly effective and eco-sustainable strategy for obtaining high-quality pomegranate seed oil with distinct biochemical signatures and increased market potential in food, nutraceutical, and cosmetic applications. Despite its advantages in yield, selectivity, and preservation of bioactives, supercritical CO_2_/IPA extraction has notable constraints. Equipment and operating costs are higher than conventional methods, and skilled personnel are required. The use of isopropanol, while improving recovery of moderately polar compounds, demands an additional removal step to ensure food-grade safety. Moreover, process parameters may slightly alter the isomeric profile of conjugated fatty acids, as observed in the Roce cultivar, potentially influencing nutritional attributes. These factors should be considered when evaluating the method’s industrial applicability. Overall, our results show that pomegranate seed oil, particularly when extracted with supercritical CO_2_/IPA, combines a unique chemical profile with bioactive and aromatic properties, making it a promising natural ingredient for innovative food products, nutraceutical formulations, and cosmetic preparations.

## Figures and Tables

**Figure 1 foods-14-02951-f001:**
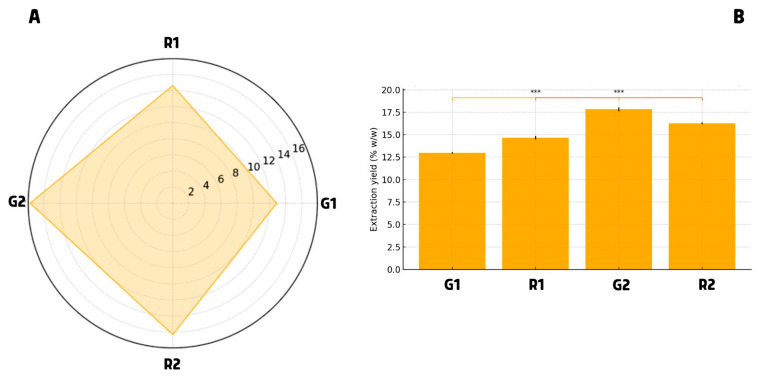
(Panel (**A**))—Radar plot of extraction yields (% *w*/*w*) of pomegranate seed oils from G1, R1, G2, and R2 samples. (Panel (**B**))—Extraction yields (% *w*/*w*) of pomegranate seed oils obtained from Granato (G1, G2) and Roce (R1, R2) cultivars using Soxhlet (G1, R1) and supercritical CO_2_/IPA (G2, R2) extraction methods. Values are expressed as mean ± SD (*n* = 3). ***Statistical differences between extraction methods for each cultivar were assessed using Student’s *t*-test (two-tailed; *p* < 0.05).

**Figure 2 foods-14-02951-f002:**
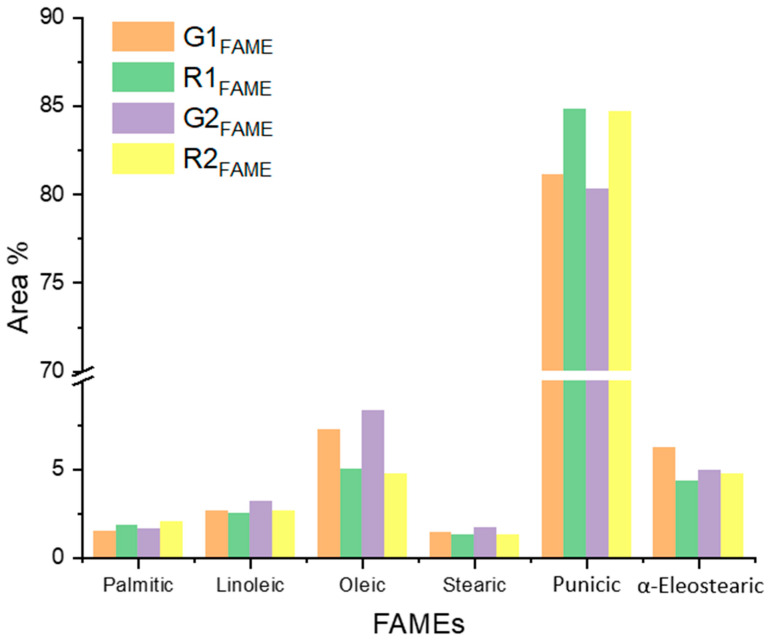
Fatty acid methyl ester (FAME) composition of pomegranate seed oil extracts from G1, G2, R1, and R2 samples. G1 and R1 refer to Soxhlet-extracted oils from *R* and *Roce* cultivars, respectively; G2 and R2 refer to oils extracted by supercritical CO_2_/IPA. Values are expressed as mean ± SD (*n* = 3).

**Figure 3 foods-14-02951-f003:**
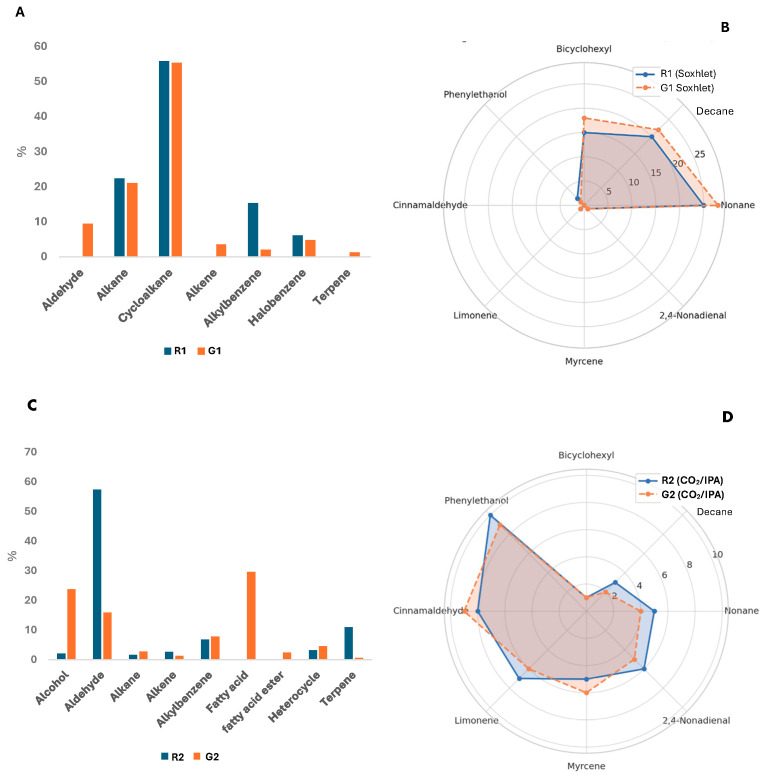
(Panel (**A**))—Distribution of the principal classes of compounds detected in composition of the G1 and R1 extracts; (Panel (**B**))—Radar plot of the volatile organic compound (VOC) profiles of G1 and R1 extracts; (Panel (**C**))—Distribution of the principal classes of compounds detected in composition of the G2 and R2 extracts; (Panel (**D**))—Radar plot of the volatile organic compound (VOC) profiles of G2 and R2 extracts.

**Figure 4 foods-14-02951-f004:**
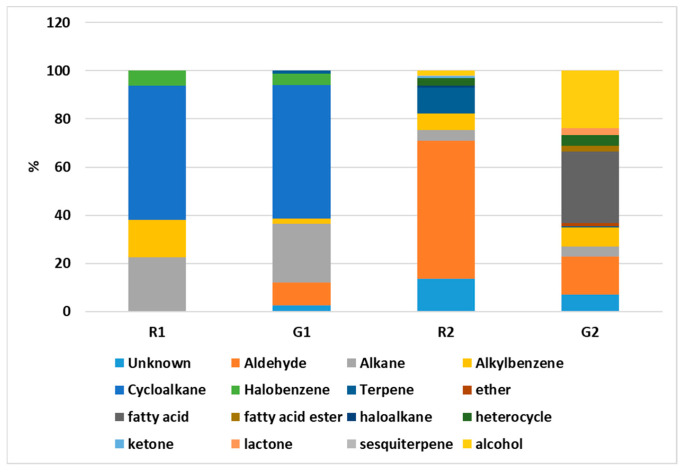
Relative contributions of volatile chemical classes in pomegranate seed oil samples. Stacked bars show the percentage area of each class for G1, R1 (Soxhlet) and G2, R2 (supercritical CO_2_/IPA). Classes include aldehydes, terpenes (mono- and sesquiterpenes), alkanes/cycloalkanes, alkylbenzenes, alcohols, esters, fatty acids, heterocycles, and others.

**Figure 5 foods-14-02951-f005:**
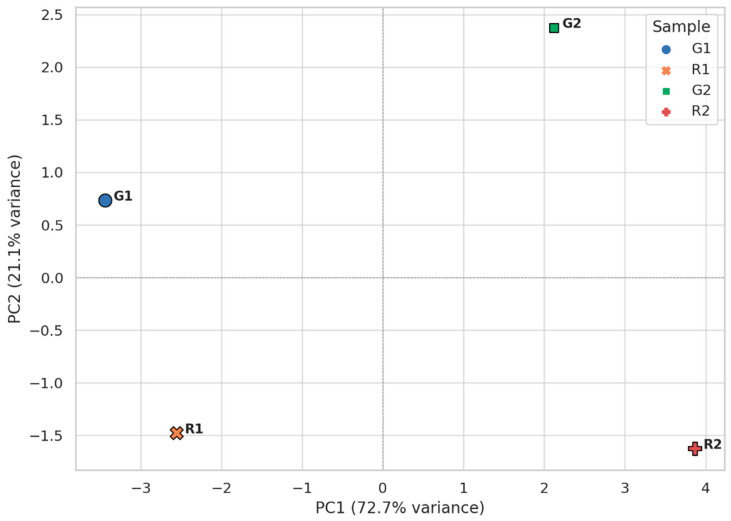
Principal Component Analysis (PCA) of pomegranate seed oil samples based on combined volatile organic compounds (VOCs) and fatty acid composition.

**Table 1 foods-14-02951-t001:** NMR characterization of the acyl profile (%) of the analyzed oil samples. Values are expressed as mean ± standard deviation (*n* = 3). Different superscript letters in the same column indicate significant differences between samples (*p* < 0.05; Tukey’s HSD test).

	Punicic ac.	α-Eleostearic ac.	Oleic ac.	Linoleic ac.	Catalpic ac.
G1_FAME_	85.5 ± 0.1 ^a^	2.8 ± 0.1 ^a^	4.5 ± 0.1 ^a^	5.8 ± 0.1 ^a^	1.4 ± 0.1 ^a^
G2_FAME_	85.0 ± 0.5 ^a^	3.0 ± 0.2 ^a^	6.0 ± 0.5 ^b^	6.0 ± 0.4 ^a^	nd *
R1_FAME_	82.2 ± 2.2 ^b^	3.3 ± 0.3 ^a^	7.6 ± 0.5 ^c^	6.9 ± 0.2 ^b^	nd *
R2_FAME_	77.5 ± 3.9 ^c^	4.7 ± 0.7 ^b^	8.6 ± 1.7 ^c^	9.2 ± 0.5 ^c^	nd *

nd *: not detectable.

**Table 2 foods-14-02951-t002:** Comparative GC-MS results obtained in the analysis of the FAMEs fraction in the G1–2_FAME_ and R1–2_FAME_ pomegranate extracts. Values are expressed as mean ± standard deviation (*n* = 3). Different superscript letters in the same column indicate significant differences between samples (*p* < 0.05; Tukey’s HSD test).

FA Precursor			Area %			
IUPAC	Common	Composition	R1_FAME_	G1_FAME_	R2_FAME_	G2_FAME_
Hexadecanoic	palmitic	C16:0	1.5 ± 0.1 ^a^	1.8 ± 0.1 ^a^	1.6 ± 0.1 ^a^	2.0 ± 0.1 ^c^
9-Cis,12-cis-octadecadienoic	linoleic	C18:2	2.6 ± 0.2 ^a^	2.5 ± 0.2 ^a^	3.2 ± 0.3 ^b^	2.6 ± 0.2 ^a^
Cis-9-octadecenoic	oleic	C18:1	7.2 ± 0.4 ^a^	5.0 ± 0.3 ^b^	8.3 ± 0.5 ^c^	4.7 ± 0.2 ^b^
Trans-9-octadecenoic	elaidic	C18:1	-	0.3 ± 0.1 ^a^	-	-
Octadecanoic	stearic	C18:0	1.4 ± 0.1 ^a^	1.3 ± 0.1 ^a^	1.7 ± 0.1 ^b^	1.3 ± 0.1 ^a^
(9Z,11E,13Z)-Octadeca-9,11,13-trienoic	punicic	C18:3	81.1 ± 1.5 ^a^	84.8 ± 0.1 ^b^	80.3 ± 1.6 ^a^	84.7 ± 1.1 ^b^
(9Z,11E,13E)-Octadeca-9,11,13-trienoic	α-eleostearic	C18:3	6.2 ± 0.3 ^b^	4.3 ± 0.2 ^a^	4.9 ± 0.3 ^a^	4.7 ± 0.2 ^a^

**Table 3 foods-14-02951-t003:** Comparative GC-MS results obtained in the analysis of the VOCs fraction in the G1 and R1 pomegranate extracts.

Compound	% Area		RI ^a^	RIL ^b^	Class of Compounds
	R1	G1			
Ethyl cyclohexane	0.67	-	835	835	Cycloalkane
Ethyl benzene	1.45	-	866	864	Alkylbenzene
Dimethyl benzene	10.11	0.94	874–875	872	Alkylbenzene
o-Xylene	2.38	-	898	891	Alkylbenzene
Nonane	1.89	3.69	904	900	Alkane
Propyl cyclohexane	0.93	-	932	931	Cycloalkane
Trans-octahydro-1H-Indene	0.65	-	957	950	Cycloalkane
4-Ethyl octane	5.96	0.36	960	966	Alkane
Cis-1-methyl-4-(1-methylethyl)-cyclohexane	0.94	-	977	984	Cycloalkane
1-Decene	-	0.44	996	990	Alkene
Mesitylene	1.43	0.51	998–999	998	Alkylbenzene
Decane	3.59	1.25	1003–1004	1000	Alkane
Limonene	-	1.27	1034	1030	Terpene
Indane	-	0.59	1042	1051	Alkylbenzene
5-(1-Methylpropyl)-nonane	-	0.39	1055	-	Alkane
Trans-decahydro-naphthalene	2.88	1.06	1057	1062	Cycloalkane
Alkane (G1: RI 1062)	0.95	5.29	1062	-	Alkane
Alkane (R1: RI 1068)	-	1.19	1068	-	Alkane
3.3-Dimethyl-hexane	-	0.55	1099	-	Alkane
Undecane	1.34	1.14	1103–1104	1100	Alkane
3.7-Dimethyl-decane	0.96	3.10	1106	1127	Alkane
Alkane (G1: RI 1115)	6.11	1.32	1115	-	Alkane
5-Ethyldecane	0.62	0.39	1149	1150	Alkane
1.2-Dichloro-4-methyl-benzene	6.15	4.76	1156	1139	Halobenzene
1-Dodecene	-	3.14	1196	1191	Alkene
Dodecane	0.49	-	1203	1200	Alkane
(2E.4Z)-2.4-Nonadienal	-	6.51	1205	1208	Aldehyde
(2E.4E)-2.4-Nonadienal	-	2.92	1227	1218	Aldehyde
Alkane (RI 1284)	0.43	1.78	1284	-	Alkane
1.1′-Bicyclohexyl	49.73	54.25	1309–1310	1298	Cycloalkane
Alkane (R1: RI 1331)	-	0.64	1331	-	Alkane
Unknown	0.34	2.50	-	-	-

^a^ Experimental retention index; ^b^ literature retention index.

**Table 4 foods-14-02951-t004:** Comparative GC-MS results obtained in the analysis of the VOCs fraction in the G2 and R2 pomegranate extracts.

Compound	% Area		RI ^a^	RIL ^b^	Class of Compounds
	R2	G2			
Octane	0.41	0.74	805	800	Alkane
2,3-Butandiol	-	3.49	814	806	Alcohol
Furfural	1.30	-	852	845	Aldehyde
Ftyrene	0.31	0.18	898	891	Alkylbenzene
2-Butyl-furan	0.34	-	899	890	Heterocycle
Nonane	0.35	0.28	905	900	Alkane
Ethyl valerate	-	0.24	913	888	Fatty acid ester
Pentanoic acid	-	25.83	945	911	Fatty acid
α-Pinene	1.93	-	938	933	Terpene
Propyl-benzene	0.65	0.86	960	962	Alkylbenzene
1-Ethyl-3-methyl-benzene	1.74	0.89	968	967	Alkylbenzene
1-Ethyl-4-methyl benzene	-	0.53	969	968	Alkylbenzene
Benzaldehyde	8.37	1.42	970/971	970	Aldehyde
1,2,4-Trimethyl-benzene	-	1.39	975	976	Alkylbenzene
β-Pinene	0.67	-	981	980	Terpene
1-Ethyl-2-methyl-benzene	0.50	0.65	986	985	Alkylbenzene
Myrcene	5.66	0.51	998	991	Terpene
Mesitylene	2.67	2.66	999	994	Alkylbenzene
Decane	0.34	0.64	1005	1000	Alkane
1,1-Diethoxy-pentane	-	1.22	1008	1016	Ether
Caproic acid	-	3.79	1019	1026	Fatty acid
1,2,3-Trimethyl-benzene	0.53	0.74	1029	1020	Alkylbenzene
p-Cymene	0.36	-	1032	1025	Terpene
Limonene	1.66	-	1035	1030	Terpene
2-Formyl-pyrrole	-	2.81	1042	1043	Heterocycle
Phenylacetaldeide	2.70	1.59	1056	1045	Aldehyde
Acetophenone	0.69	-	1080	1068	Ketone
Glycerol	-	16.29	1085	-	Alcohol
2-Ethyl-1,3-dimethyl-benzene	0.42	-	1092	-	Alkylbenzene
Terpinolene	0.68	-	1095	1086	Terpene
Methylbutyl-2- isovalerate	-	1.02	1107	1110	Fatty acid ester
2-Hydroxy-ethyl hexanoate	-	1.23	1109	1098	Fatty acid ester
Nonanal	0.62	0.99	1113	1102	Aldehyde
Thiazole	2.89	1.02	1121/1122	-	Heterocycle
5-Methyl-6,7-dihydro-5(H)-cyclopentapyrazine	-	0.71	1123	1139	Heterocycle
Phenethyl alcohol	2.11	3.86	1129	1113	Alcohol
1-Bromo-octane	0.64	-	1149	-	Haloalkane
Dehydromevalonic lactone	-	2.86	1184	-	lactone
1-Dodecene	1.93	0.99	1197	1191	Alkene
(2E,4Z)-2,4-Nonadienal	6.57	7.12	1204	1196	Aldehyde
(2E,4E)-2,4-Nonadienal	4.98	2.81	1225	1218	Aldehyde
(E)-Cinnamaldehyde	32.89	1.96	1286/1287	1273	Aldehyde
Tridecane	0.40	0.52	1305	1300	Alkane
Tetradecene	0.74	0.32	1397/1398	1392	Alkene
Tetradecane	0.19	0.30	1405	1400	Alkane
E-Cariophyllene	0.22	-	1432	1424	Sesquiterpene
Pentadecane	-	0.18	1505	1500	Alkane
β-Bisabolene	-	0.11	1521	1511	Terpene
2,6-Bis(1,1-dimethylethyl)-4-methyl-phenol	-	0.13	1526	1514	Alcohol
Hexadecane	-	0.09	1606	1600	Alkane
Unknown	13.53	7.02	-	-	

^a^ Experimental retention index; ^b^ literature retention index.

## Data Availability

The original contributions presented in this study are included in the article/Appendix A. Further inquiries can be directed to the corresponding author.

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
