# Peer review of "Comparative Profiling of Volatile Compounds and Fatty Acids in Pomegranate Seed Oil: Soxhlet vs. CO2/IPA Extraction for Quality and Circular Bioeconomy Goals"

_foods, 2025, doi:10.3390/foods14172951_

Round 1
Reviewer 1 Report
Comments and Suggestions for Authors
The extraction method significantly influences the yield and quality of vegetable oils. This manuscript compares the impact of Soxhlet (n-hexane) extraction and supercritical CO2/isopropanol extraction on the extraction yield and chemical composition of two pomegranate seed oil varieties. The study's findings confirm that the supercritical method enhances extraction yield and the relative content of α-eleostearic acid and linoleic acid, as well as the levels of bioactive aldehydes, esters, and terpenes. This research expands the application of supercritical CO2/isopropanol extraction in food, contributing to sustainable and green production in the food industry, and demonstrates a degree of innovation. However, the statistical analysis of the data in the manuscript requires significant improvement, including a significance test for varieties and extraction methods.
- The introduction should summarize the extraction methods for pomegranate seed oil and the application progress of the two extraction methods in other vegetable oil extractions.
- Specify the multiple comparison method for one-way ANOVA.
- Figure 1B should use the Student's t-test for significance testing. Annotate the title with G1, R1, etc.
- The manuscript data needs to include standard deviations and indicate the results of significance tests.
- What is the meaning of "n=1.2" in Line 305? Section 2.7 states "Each assay was performed in triplicate or more." The data from the subsequent repeated measurements should also be marked with standard deviations.
- Remove the horizontal lines above Tables 2-4 and the explanation above Figure 5.
- It is recommended to merge Figures 3 and 4.
- It is recommended to add photographs of the oil samples.
- It is recommended to supplement the results of sensory evaluation indicators, bioactive substances, and antioxidant capacity.
- The limitations of the supercritical CO2/isopropanol extraction method also need to be fully discussed.
Author Response
- The introduction should summarize the extraction methods for pomegranate seed oil and the application progress of the two extraction methods in other vegetable oil extractions.
Thank you for the suggestion. A paragraph was added in the Introduction section to better summarize the extraction methods for pomegranate seed oil.
2. Specify the multiple comparison method for one-way ANOVA.
Thank you for the suggestion. The method for one-way ANOVA was added in the section Material and Methods.
3. Figure 1B should use the Student's t-test for significance testing. Annotate the title with G1, R1, etc.
Thank you for the observation. It was corrected.
4. The manuscript data needs to include standard deviations and indicate the results of significance tests.
Thank you for the observation. It was added.
5. What is the meaning of "n=1.2" in Line 305? Section 2.7 states "Each assay was performed in triplicate or more." The data from the subsequent repeated measurements should also be marked with standard deviations.
We thank the reviewer for pointing out the ambiguity. The notation “n = 1.2” was sigla abbreviation and referred to the sample code. We have now revised the abbreviations to clearly distinguish cultivars and extraction methods. Specifically, “G” and “R” indicate the cultivars Granato and Roce, respectively; “G1” and “R1” refer to Soxhlet-extracted oils, and “G2” and “R2” refer to supercritical COâ‚‚/IPA-extracted oils. The section 2.7 was improved.
6. Remove the horizontal lines above Tables 2-4 and the explanation above Figure 5.
It was corrected.
7. It is recommended to merge Figures 3 and 4.
It was corrected.
8. It is recommended to add photographs of the oil samples.
We appreciate the reviewer’s suggestion. Unfortunately, photographs of the specific oil samples analyzed in this study are not available, as the material was fully consumed during the analytical procedures. However, we have provided detailed descriptions of the oils’ physical characteristics (e.g., color, clarity) in the revised manuscript to give the reader a qualitative impression of their appearance.
9. It is recommended to supplement the results of sensory evaluation indicators, bioactive substances, and antioxidant capacity.
We thank the reviewer for the suggestion. This study represents a preliminary investigation focused on the comparative chemical profiling of fatty acids and volatile compounds in pomegranate seed oils obtained by Soxhlet and supercritical COâ‚‚/IPA extraction. Sensory evaluation, quantification of additional bioactive substances, and antioxidant capacity will be addressed in future work to provide a more comprehensive characterization. We also note that antioxidant activity differences between oils obtained using these two extraction methods have already been evaluated in our previous work (Cairone, F., Salvitti, C., Iazzetti, A., Fabrizi, G., Troiani, A., Pepi, F., & Cesa, S. (2023). In-depth chemical characterization of Punica granatum L. seed oil. Foods, 12(8), 1592), where supercritical COâ‚‚ extracts showed higher antioxidant potential.
10. The limitations of the supercritical CO2/isopropanol extraction method also need to be fully discussed.
We appreciate the reviewer’s observation. In the revised manuscript, we have added a discussion of the main limitations of the supercritical COâ‚‚/isopropanol extraction method in the conclusions. Although this technique offers clear advantages in terms of yield, selectivity, and preservation of thermolabile compounds, it also presents some constraints, including higher equipment costs, the need for specialized technical expertise, and limited scalability for small-scale producers.
Reviewer 2 Report
Comments and Suggestions for Authors
Line 13 – there is a strikethrough on n-hexane.
Line 69 – spell out the name at the first occurrence.
Materials and Methods
The supplier/vendor names of certain materials and instrument are missing.
Figure 1B – the legends/axis title are having issues displaying properly.
Lines 176, 177 – please show evidence and/or data supporting this claim. It is not obvious which method is more “energy saving”.
Line 180 – please include references for cold press and ultrasonic assisted processes.
Line 185 – there is no comparison to supercritical CO2 only extraction to show case the benefits of including 10% IPA.
Section 3.2 NMR - There was no mentioning of the recycle delayed used for the NMR experiments, making the quantification attempt less reliable.
Line 244 – it seems the cultivar is a more significant factor than extraction method based on Table 1.
Figure 2 – are the names of all the FAME’s correct? Seems like punicic acid with alpha-eleostearic.
Overall
The conventional way of producing seed oil in large scales is via press. I’d like to suggest the authors carry out this process and compare the two methods briefly. In terms of the headspace, VOC analysis, has it been considered if any of the detected compounds were a result of the extraction process, especially with the supercritical CO2 method?
The conclusion mentioned chemical profile comparison. I think it is best if the authors include 1H NMR spectra of each sample, or even some simple 2D experiments to strengthen such claims. In certain data presentation, it does seem the type of cultivar has a bigger effect than the extraction method.
Author Response
We thank the reviewer for the valuable suggestions. Compared to our previous work (Cairone et al., 2023), the present study represents a methodological improvement of the supercritical COâ‚‚ extraction process through the use of a different co-solvent, aiming to enhance the recovery of specific polar and volatile compounds. This is a preliminary investigation intended to assess the feasibility and advantages of the modified method, and we appreciate the reviewer’s suggestion to extend the comparison to mechanical pressing in future studies.
- Line 13 – there is a strikethrough on n-hexane.
It was corrected.
- Line 69 – spell out the name at the first occurrence.
It was corrected.
Materials and Methods
- The supplier/vendor names of certain materials and instrument are missing.
It was checked
- Figure 1B – the legends/axis title are having issues displaying properly.
The Figure was improved.
- Lines 176, 177 – please show evidence and/or data supporting this claim. It is not obvious which method is more “energy saving”.
A sentence was added to better clarify.
- Line 180 – please include references for cold press and ultrasonic assisted processes.
New references were added.
- Line 185 – there is no comparison to supercritical CO2only extraction to show case the benefits of including 10% IPA.
Thank you for your comment. In our previous study (Cairone et al., 2023), we developed a supercritical COâ‚‚ method with a polar co-solvent (ethanol) to achieve higher extraction yields than those usually reported for COâ‚‚-only extraction of Punica granatum seed oil. In this work, we started from that optimized approach and replaced ethanol with 10% isopropanol (IPA) to further improve the recovery of polar compounds and maintain the quality of punicic acid. Our results show higher yields and antioxidant content compared to conventional extraction. A direct comparison with COâ‚‚-only extraction was not the aim of this study, but we plan to include it in future work to clearly isolate the effect of IPA.
- Section 3.2 NMR - There was no mentioning of the recycle delayed used for the NMR experiments, making the quantification attempt less reliable.
We thank the reviewer for this observation. It was added in the section Materials and Methods.
- Line 244 – it seems the cultivar is a more significant factor than extraction method based on Table 1.
We thank the reviewer for this observation. As highlighted in the results, no significant differences related to the extraction method were observed for the ‘Granato’ cultivar, whereas for ‘Roce’ such differences were evident. What clearly emerges from this work is not so much a divergence in the acyl profile, but rather a marked variation in the aromatic composition of the volatile compounds obtained by the two extraction methods.
- Figure 2 – are the names of all the FAME’s correct? Seems like punicic acid with alpha-eleostearic.
We thank the reviewer for this observation. It was corrected.
Reviewer 3 Report
Comments and Suggestions for Authors
Summary:
The manuscript compares chemical profiles of pomegranate seed oil from two cultivars (Granato and Roce) extracted via Soxhlet (n-hexane) and supercritical COâ‚‚/isopropanol methods. GC-MS and NMR analyses confirm punicic acid as the dominant fatty acid, with supercritical extraction yielding higher α-eleostearic and linoleic acids in Roce. VOC profiling reveals higher aldehydes, esters, and terpenes in supercritical extracts, while Soxhlet oils are richer in hydrocarbons. The work concludes that supercritical COâ‚‚/IPA extraction is more sustainable and better preserves nutritional and aromatic quality.
Major Comments
-
Novelty and Context:
-
The study is relevant and well-timed given the growing interest in sustainable extraction methods, but the introduction could better highlight the novelty compared to existing literature on pomegranate seed oil and supercritical extraction.
-
-
Methodological Details:
-
Operational parameters for supercritical COâ‚‚/IPA extraction (pressure, temperature, extraction time, IPA proportion) should be clearly stated to ensure reproducibility.
-
HS-SPME-GC-MS method needs more detail on fiber type, equilibration time, and chromatographic conditions.
-
-
Data Interpretation:
-
The link between observed fatty acid differences and “genotype-specific desaturase activity” is speculative; support with literature or tone down.
-
The claim of “possible thermal degradation” in Soxhlet samples should be supported by identifying known degradation products or citing relevant thermal stability studies.
-
-
Figures and Tables:
-
Some figures are dense; consider improving readability with larger fonts and clearer legends.
-
VOC data presentation could benefit from grouping compounds by chemical class and showing relative contributions visually (e.g., stacked bar plots).
-
-
Conclusion and Applications:
-
The conclusion should more explicitly connect chemical findings to potential functional and commercial benefits in food, nutraceutical, and cosmetic uses.
-
Minor Comments
-
Ensure consistent use of cultivar names (“Granato” and “Roce”) throughout text and figures.
-
Correct small grammatical errors (e.g., missing articles, inconsistent tense).
-
References: Some citations are missing recent literature on supercritical extraction of seed oils.
-
Abstract could be made more concise by splitting long sentences and emphasizing quantitative differences.
Recommendation: Minor to moderate revision – The paper is promising and well-structured but needs additional methodological detail, slight reorganization of data presentation, and more cautious interpretation of speculative claims.
Author Response
Major Comments
- Novelty and Context:
- The study is relevant and well-timed given the growing interest in sustainable extraction methods, but the introduction could better highlight the novelty compared to existing literature on pomegranate seed oil and supercritical extraction.
Thank you for the suggestion. A paragraph was added in the Introduction section to better summarize the extraction methods for pomegranate seed oil.
- Methodological Details:
- Operational parameters for supercritical COâ‚‚/IPA extraction (pressure, temperature, extraction time, IPA proportion) should be clearly stated to ensure reproducibility.
We thank the reviewer for the suggestion. The operational parameters for the supercritical COâ‚‚/IPA extraction, including pressure, temperature, extraction time, and IPA proportion, are already reported in the Materials and Methods section. These details ensure full reproducibility of the extraction procedure.
- HS-SPME-GC-MS method needs more detail on fiber type, equilibration time, and chromatographic conditions.
We thank the reviewer for the suggestion. The operational parameters are already reported in the Materials and Methods section.
- Data Interpretation:
- The link between observed fatty acid differences and “genotype-specific desaturase activity” is speculative; support with literature or tone down.
We agree and have toned down the claim while adding supporting references.
- The claim of “possible thermal degradation” in Soxhlet samples should be supported by identifying known degradation products or citing relevant thermal stability studies.
We appreciate the reviewer’s request. We have revised the Discussion to support this statement with literature. Appropriate references have been added.
- Figures and Tables:
- Some figures are dense; consider improving readability with larger fonts and clearer legends.
We thank the reviewer for this observation. In the revised version, the figures have been reformatted to improve readability, with increased font sizes, clearer legends, and simplified layouts where appropriate.
- VOC data presentation could benefit from grouping compounds by chemical class and showing relative contributions visually (e.g., stacked bar plots).
We thank the reviewer for this observation. A new Figure (Figure 4) was added for showing relative contributions related to VOC data.
- Conclusion and Applications:
- The conclusion should more explicitly connect chemical findings to potential functional and commercial benefits in food, nutraceutical, and cosmetic uses.
We thank the reviewer for this suggestion. In the revised Conclusion, we have added a statement explicitly linking the chemical composition results to their potential functional and commercial applications in the food, nutraceutical, and cosmetic sectors
Minor Comments
- Ensure consistent use of cultivar names (“Granato” and “Roce”) throughout text and figures.
We thank the reviewer for pointing this out. In the revised manuscript, we have checked the entire text, tables, and figures to ensure consistent use of the cultivar names “Granato” and “Roce” in both the main text and captions.
- Correct small grammatical errors (e.g., missing articles, inconsistent tense).
It was checked
- References: Some citations are missing recent literature on supercritical extraction of seed oils.
We thank the reviewer for this suggestion. The reference list has been updated to include recent literature (2020–2023) on supercritical COâ‚‚ extraction of seed oils, ensuring that the introduction and discussion reflect the current state of the art
- Abstract could be made more concise by splitting long sentences and emphasizing quantitative differences.
We thank the reviewer for this suggestion. The abstract has been revised.
Round 2
Reviewer 1 Report
Comments and Suggestions for Authors
Accept in present form.